# Design of a Greenhouse Solar-Assisted Heat Pump Dryer for Kelp (*Laminaria japonica*): System Performance and Drying Kinetics

**DOI:** 10.3390/foods11213509

**Published:** 2022-11-04

**Authors:** Huanyu Kang, Guochen Zhang, Gang Mu, Cheng Zhao, Haolin Huang, Chengxiang Kang, Xiuchen Li, Qian Zhang

**Affiliations:** 1College of Mechanical and Power Engineering, Dalian Ocean University, Dalian 116023, China; 2Technology Innovation Center of Marine Fishery Equipment in Liaoning, Dalian 116023, China; 3Key Laboratory of Environment Controlled Aquaculture Ministry of Education, Dalian Ocean University, Dalian 116023, China

**Keywords:** solar drying, heat pump, design, kelp, system performance, drying kinetics

## Abstract

In order to solve a series of problems with kelp drying including long drying time, high energy consumption, low drying efficiency, and poor quality of dried kelp, this work proposes the design of a novel greenhouse double-evaporator solar-assisted heat pump drying system. Experiments on kelp solar-assisted heat pump drying (S−HP) and heat pump drying (HP) under the condition of irradiance of 100−700 W/m^2^ and a temperature of 30, 40, or 50 °C were conducted and their results were compared in terms of system performance, drying kinetics, and quality impact. The drying time was reduced with increasing irradiance or temperature. The coefficient of performance (COP) and specific moisture extraction rate (SMER) of S−HP were 3.590−6.810, and 1.660−3.725 kg/kW·h, respectively, roughly double those of HP when the temperatures are identical. The *Deff* of S-HP and HP were 5.431 × 10^−11^~11.316 × 10^−11^ m^2^/s, and 1.037 × 10^−11^~1.432 × 10^−11^ m^2^/s, respectively; additionally, solar radiation greatly improves *Deff*. The Page model almost perfectly described the changes in the moisture ratio of kelp by S−HP and HP with an inaccuracy of less than 5%. When the temperature was 40 °C and the irradiance was above 400 W/m^2^, the drying time of S−HP was only 3 h, and the dried kelp maintained the green color with a strong flavor and richness in mannitol. Meanwhile, the coefficient of performance was 6.810, the specific moisture extraction rate was 3.725 kg/kWh, and the energy consumption was 45.2%, lower than that of HP. It can be concluded that S−HP is highly efficient and energy-saving for macroalgae drying and can serve as an alternate technique for the drying of other aquatic products.

## 1. Introduction

Kelp (*Laminaria japonica*) [1,2] is a valuable, affordable and nutrient-rich marine resource. Due to its industrial and edible value, as well as its medicinal functions, such as decreasing blood pressure, blood lipids, and blood sugar, and alleviating radiation and goiter [2,3], global kelp production was increased to 1.936 million tons in 2021 [4]. The moisture content of fresh kelp is above 90%, which makes it extremely perishable. Therefore, it is preferable to process fresh kelp into dried kelp [5], which has high added value and a long shelf life, and the global market demand for dried kelp has also witnessed continuous increases [6].

The traditional drying method (natural drying) is slow and the quality of its products is unstable [7]. Additionally, for the majority of other drying techniques, excessive energy consumption is always a problem in view of the global energy landscape. Therefore, it is imperative for the kelp industry to locate novel drying methods that can achieve energy conservation while maintaining high efficiency. One of the most widely used energy-efficient methods is heat pump drying (HP) [8]. This method transfers heat from low-temperature media to high-temperature media, which can reduce energy consumption and improve the quality of its products. Zhang et al. [9] studied the kinetic and thermodynamic characteristics of HP for kelp and discovered that temperature, wind speed, humidity, and kelp thickness all affected the kinetic characteristics of HP. In the study, the ideal condition for kelp drying was a temperature of 40 °C, a wind speed of 1.3 m/s, and a humidity of 40%, without considering the heat pump system’s performance. Hu et al. [10] examined a closed-loop double-evaporator heat pump dryer (HPD) used for kelp knots and found that the startup time of the instrument was 20.8% less than a conventional single-evaporator HPD, while its initial investment was only 6.5% more. However, the closed HP method faced serious problems caused by high relative humidity during kelp drying, which prolonged the drying time, increased the energy consumption and shortened the life of the instrument.

In order to further cut down on energy consumption and cost, scholars have continued to make efforts to improve drying techniques. Zhong et al. [11], Koan et al. [12] and Singh et al. [13] explored heat-collection solar-assisted heat pump drying (S−HP), which used solar energy collectors to accumulate high densities of energy to heat drying media (air, water, oil, etc.) to a set temperature for direct or indirect material dehydration. Zhong et al. [11] studied the drying characteristics of Chinese wolfberry and the thermal efficiency of its drying at 40−70 °C using indirect solar collectors in conjunction with HPDs. It was observed that the solar-assisted heat pump drying system (SHPD) used less energy than the HPD alone, but because the drying process involved high relative humidity, an additional dehumidifier was required. Koan et al. [12] created a novel type of photovoltaic/thermal solar collector system for mint leaf drying, and the system’s heating, drying, and power supply capacities were tested. Variation in solar irradiances was found to be able to affect the evaporator’s thermal expansion and cause system instability. Singh et al. [13] compared the efficiency of HP and heat-collection S−HP for banana slices at a drying temperature of 60 °C and discovered that the S−HP consumed 31% less overall energy while the coefficient of performance (COP) was increased by 28%. Moreover, a previous study on heat-collection S−HP manifested its energy-saving potential at a high-temperature range of 60−70 ℃ [14]. In contrast, some other studies found that the ideal drying temperature range for kelp was 30−50 °C, a temperature range for low-temperature drying. Furthermore, the significant quantity, the high moisture content and the construction complexity should also be taken into consideration for the design of SHPD.

In this context, this work proposes the design of a greenhouse double-evaporator SHPD to meet the needs of batch drying and processing for kelp and other aquatic products in bulk, and to avoid heat and humidity mismatch, a common problem of HP. In addition, the system performance and drying kinetics of the SHPD for kelp were examined with the aim of providing theoretical and technical support for the future development of energy-saving and high-efficiency drying technology and equipment for bulk low-value aquatic products.

## 2. Materials and Methods

### 2.1. SHPD

The proposed SHPD combines a double-evaporator heat pump with the greenhouse solar drying technology, the design is different from the familiar solar collector combined heat pump dryer reported by Yahya et al. [15], Qui et al. [16] and Mohanraj et al. [17]. Figure 1 shows the schematic of the SHPD, which consists of a heat pump system (including a compressor, a condenser, an evaporator, an evaporative condenser, an expansion valve, and the refrigerant R134a) and a solar energy system (including a solar drying chamber, a fan, an air duct, and air valves). The specifications of the key components are shown in Table 1.

The system is equipped with three air circulation modes, i.e., open-loop, closed, and semi-open, and two operation modes, i.e., single condensation/single evaporation and single condensation/double evaporation. Single condensation/single evaporation is a conventional heat pump structure, but the control of humidity in the drying chamber is not ideal. Therefore, the mode of single condensation/double evaporation mode was designed to rapidly improve the temperature and effectively regulate the humidity in the drying chamber, so as to achieve the balance of heat and humidity during drying. By adjusting the air valves or controlling the compressor, the specified temperature of the drying chamber can be reached, and the temperature has continuously been monitored by an electronic thermostat throughout the drying process, which sends real-time data to the control panel. To regulate the temperature and humidity in the dryer, the area of the evaporator is changed by altering the number of operating evaporators.

HP mode: The single evaporation-single condensation mode is the first option based on energy-saving considerations. The dehumidifying evaporator is opened, and the solenoid valve, the fresh air valve and the exhaust air valve are closed. The hot humid air expelled from the drying chamber is dried by the dehumidifying evaporator and heated by the condenser, before it enters the drying chamber again to achieve a closed cycle. However, when the humidity is too high, the single condensing-double evaporator mode will be turned on by an intelligent control system. The solenoid valve is opened, so the assistant evaporator starts to absorb heat from the outside air, and the dehumidifying evaporator starts to absorb heat from the air in the chamber. The dried air flows through the condenser and is heated up, then enters the drying chamber to complete the semi-open circulation.

S−HP mode: Through this mode, solar energy is utilized to heat up the drying chamber, and the heat pump drying system is only turned on when the temperature cannot reach the desired value. When the drying temperature is 10 °C higher than the set temperature, the fresh air valve and the exhaust air valve is opened, and the middle air valve is closed, so as to implement the open-loop circulation.

### 2.2. Materials and Drying Procedure

Fresh kelp was harvested from Lvshun, Dalian in June 2021. Medium-sized kelp slices of 167.7 ± 4.3 cm in length, 30.0 ± 5.0 cm in breadth, and 1.1 ± 0.15 kg in weight were chosen and divided into 27 groups (each with 18 slices and a total weight of 20 ± 3 kg). The average outdoor temperature and relative humidity were 28.6 ± 2.1 °C, and 62.1 ± 6.7%, respectively. The kelp slices were flattened with a fixture and hung vertically on the drying rack in a 6-row × 3-column pattern (Figure 2). The initial moisture content of the fresh kelp was determined as 86.3% in a drying oven (BPJ-9123-A, Shanghai Instrument Manufacturing Co., Ltd., Shanghai, China) at 105 °C. The roller curtain for thermal insulation was rolled up in S−HP mode and rolled down in HP mode. This process was completed by the control system. After preheating, the drying rack was pushed into the drying chamber. Data were recorded every 1 h until the set moisture content of 18% on wet basis (w.b.) was reached. The arrangement of the drying tests was showing in Table 2.

### 2.3. Drying Kinetics

First, the moisture ratio (*MR*) was calculated as follows:(1)$$M_{wb}=\frac{m_{t}-m_{0}\left(1-M_{0}\right)}{m_{t}}\times 100\%$$
(2)$$M_{db}=\frac{M_{wb}}{1-M_{wb}}\times 100\%$$
(3)$$MR=\frac{M_{t}-M_{e}}{M_{0}-M_{e}}$$
where *M_wb_* is the wet basis moisture content over a particular drying time *t* (g H_2_O/g w.b.), *M_db_* is the dry basis moisture content over a particular drying time *t* (g H_2_O/g d.b.), *m_t_* is the sample weight over a particular drying time *t* (g), *m*_0_ is the initial weight of the sample (g), *M*_0_ is the initial moisture content (g H_2_O/g w.b), *M_t_* is the moisture content at time *t* and *M_e_* is the equilibrium moisture content (g H_2_O/g d.b).

Formula (3) could be simplified [18] as:(4)$$MR=M_{t}/M_{0}$$

Then, the widely used Henderson-Pabis, Page, Wang and Sing models, the Lewis for thin-layer drying were adopted and linearization was conducted on each model (Table 3).

### 2.4. Effective Moisture Diffusivity

Effective moisture diffusivity (*Deff*) was calculated according to Fick’s Second Law:(5)$$lnMR=ln\frac{8}{\pi^{2}}-\frac{\pi^{2}D_{eff}}{L^{2}}t$$
where *Deff* is the effective moisture diffusivity of the material (m^2^/s), *L* is the thickness of the material (m) and *t* is the drying time (s).

### 2.5. Sensory Properties and Texture

The dried kelp was evaluated referring to the national standard specification for dried kelp SC/T 3202-2012 in terms of the factors of color, shape, flavor, and shrinkage. A centesimal system was adopted as shown in Table 4, and 5 slices of kelp were sampled from each group for evaluation. Five panel members trained in food sensory evaluation were employed to score their evaluation of all groups. The average value of the scores for a group was used as the final score of the group.

The top, main and tail parts of the rehydrated kelp were sampled. The specimens were cut into 1 × 1 cm pieces and analyzed for texture profile analysis (TPA) using a TA/36R probe. The initial force was set to 0.1 N, the loading rate was 10 mm/min, the height from the probe to a specimen was 100 mm, and the textural indices included hardness, springiness, and chewiness. The deformation percentage was 50% (n = 5) [23].

### 2.6. Energy Consumption

An electric energy meter was used to measure energy consumption. The specific moisture extraction rate (SMER) was calculated as follows:(6)$$\mathrm{SMER}=\frac{M_{d}}{W}$$
where SMER is the specific moisture extraction rate (kg/kW·h), *M_d_* is the moisture evaporation (kg) and *W* is the electric energy (kW/h).

The COP was calculated as follows:(7)$$\mathrm{COP}=1+\mathrm{SMER}\times h_{tg}$$
where *h_tg_* is the latent heat of the evaporated water, which is 1.56 kg/(kW·h) at 100 °C.

### 2.7. Statistical Analysis

The model parameters were determined using the Levenberg Marquardt algorithm for nonlinear analysis in IBM SPSS statistics 26 (IBM SPSS lnc., Chicago, IL, USA). The coefficient of determination (R^2^), the root mean square error (RMSE), and the Chi square (ꭓ^2^) were calculated as follows:(8)$${\;R}^{2}=1-\frac{\sum_{1}^{N}{\left(MR_{exp,i}-MR_{pre,i}\right)}^{2}}{\sum_{1}^{N}{\left({\bar{MR}}_{exp}-MR_{pre,i}\right)}^{2}}$$
(9)RMSE=[1N∑1N(MRexp,i−MRpre,i)2]12 
(10)$${\;\chi }^{2}=\frac{\sum_{1}^{N}{\left(MR_{exp,i}-MR_{pre,i}\right)}^{2}}{N-n}$$
where *MP_pre,i_* is the predicted value of the *MR*, *MR_exp,i_* is the experimental value of the *MR*, *N* is the number of observations, and *n* is the number of constant terms in the regression model.

The data were processed by Microsoft Office Excel 2016 (Microsoft Inc., Redmond, Washington, DC, USA), and the analysis of variance (ANOVA) was performed in the IBM SPSS Statistics 26 software. Principal component analysis (PCA) was completed in Origin pro 2021b (Origin Lab Corporation, Northampton, MA, USA) using Principal Component Analysis and Correlation plot.

## 3. Results and Discussion

### 3.1. Effect of S−HP on Dehydration Characteristics of Kelp

Variations in MRs and drying rates under the conditions of different irradiances (400 W/m^2^, 200−400 W/m^2^, or 200 W/m^2^) and drying temperatures (30, 40, or 50 °C) in S−HP mode are shown in Figure 3. The moisture content was reduced to 18% (w.b.) in 3−5 h and increases in irradiance or drying temperature were found to be conducive to the reduction in drying time. Compared with that of S−HP ≥ 400 W/m^2^, the drying time of S−HP 200−400 W/m^2^ and S−HP ≤ 200 W/m^2^ was prolonged by 33%. There was no significant difference between the drying rate at 50 °C and that at 40 °C (*p* < 0.05), but both were significantly higher than that at 30 °C. This is similar to the results of the research by Moot et al. [24] on solar drying for alfalfa. The penetration depth of strong and weak solar irradiation differed, leading to a nearly 10 times difference in the drying rate.

### 3.2. Effect of HP on Dehydration Characteristics of Kelp

Variations in MRs and drying rates in HP mode under the conditions of different drying temperatures (30, 40, or 50 °C) are shown in Figure 4. The drying time of kelp HP was between 4−5 h and would be reduced as the temperature increased. The drying time was 20% shorter at 40 °C and 50 °C than at 30 °C. The drying rate was relatively high at the early drying stage and positively regulated by the temperature. Moreover, the drying rate decreased as the moisture content decreased, and the limiting factor changed from surface diffusion control to internal migration control, so there was almost no difference in drying rate at different temperatures at the late drying stage [25]. There was no obvious constant-speed stage during the kelp drying. This may be due to the rapid evaporation on the surface of kelp, which makes it difficult to form a stable water concentration difference between the surface and the interior of kelp [26]. The results were consistent with the pattern observed by Zhang et al. [9] in the HP for small-sized kelp slices, though the drying time of the proposed SHPD was shortened by 2–7 h. In addition, when the temperature was 40−50 °C, the drying time of HP was 33% longer than that of S−HP, and the average drying rate was 7.3% lower, indicating that solar irradiation could significantly improve the drying rate of kelp.

### 3.3. Determination of the Drying Model

Multi-heat transfer drying is a complex process of heat and mass transfer. The MRs of kelp drying under different drying conditions were fitted to the above-mentioned three models (Table 5). The results show that the correlation coefficients of the Page model ranged from 0.873 to 0.999, and the average values of RMSE and χ^2^ (0.06, 0.02, respectively) were closer to 0, indicating that the Page model was better than the other three models in describing the changing process of the moisture content.

Figure 5 presents the residual distributions for all groups (total experimental points = 53). The residual distribution of the Henson−Pabis model (Figure 5a) had a V−pattern. The residual distribution of the Wang and Sing model (Figure 5c) was highly clustered and the residual distribution of the Page model (Figure 5b) and the Lewis model (Figure 5d) ranged from −1.864 to 2.135 and −2.953 to 4.893 with good dispersion. This indicated that the Page model could describe the drying kinetics of kelp more accurately. In a previous study on the HP for small-sized kelp slices, the Page model also demonstrated effective fitting to the variations in MRs [9], but Samimi et al. [27] and Roa et al. [28] found that the Aghbashlo model and the Midilli−Kucuk model were better in describing solar hot air drying of tomato slices and in solar drying of cherries.

The predicted and experimental values of S−HP ≥ 400 W/m^2^ were compared, as shown in Figure 6, to verify the accuracy of the model. The predicted curve fits well with the experimental value with an error that is less than 5%.

### 3.4. Deff

The Deff under different conditions is shown in Table 6, which varies from 1.037 × 10^−11^ m^2^/s to 11.316 × 10^−^^11^ m^2^/s, within reasonable bounds for biological materials [29]. Kelp’s Deff in S−HP mode ranged from 5.431 × 10^−^^11^ m^2^/s to 11.316 × 10^−^^11^ m^2^/s, which was seven times that in HP mode. In particular, the Deff of S−HP ≥ 400 W/m^2^ was 80.8% higher than that of S−HP ≤ 200 W/m^2^. This was because the absorbed solar energy was transformed into the internal energy of the kelp, which expedited the moisture migration and diffusion [30].

When the temperature increased from 30 °C to 50 °C, the Deff increased by 38.1% from 1.037 × 10^−^^11^ m^2^/s to 1.432 × 10^−^^11^ m^2^/s in HP mode and increased by 92.0% from 5.431 × 10^−^^11^ m^2^/s to 10.428 × 10^−^^11^ m^2^/s in S−HP mode. This result indicates that temperature had a positive effect on the Deff, and solar irradiation intensified the effect. On the one hand, the higher temperature not only resulted in a higher moisture evaporation rate on the surface of the kelp [31] but also decreased the internal moisture viscosity, which was conducive to the internal moisture diffusion [32]. On the other hand, the introduction of solar irradiation facilitated energy penetration through the surface to realize direct heating for the interior. Since the thickness of kelp slices could significantly change the activation energy and enthalpy of the kelp [9], the thermal radiation in S−HP mode made up for the defect that the heat convection energy provided by the simple HP mode cannot quickly penetrate into the interior of kelp, and the efficient heat transfer contributed to the rapid moisture migration.

### 3.5. Quality of the Dried Kelp

Table 7 illustrates the texture and sensory properties of the dried kelp under various drying conditions. The relationship between the textural indices and the sensory score of kelp can be modeled by a quadratic function [33]. By comparing sensory scores and texture data, it can be seen that the sensory evaluation improves with increasing hardness and springiness, and that chewiness within the range of 50−90 mJ was preferred. After drying, rehydrated kelp showed a superior texture to the raw material. The HP groups had higher hardness, while the S−HP groups showed better springiness and chewiness on average, as well as better consistency in texture quality. The average sensory score of kelp was 77.87, with little difference between the groups. The dried kelp was in uniform bright green with a strong natural kelp flavor, and the overall shape was relatively complete. This is consistent with the results obtained by Pierrick et al. [1] in the study on the kelp color changes during hot air drying at 40−70 °C. He found that the kelp was green at 40 °C, and it turned yellow due to the exposure of carotenoids under high drying temperatures. Mannitol was precipitated on the surface of the kelp, and the edge of the kelp turned slightly yellow. The sensory quality deteriorated as the drying temperature increased. This was because the lipases on the thylakoid membranes of kelp were more activated as the temperature was higher to degrade the cell membrane lipids and altered the internal structure of kelp cells [34]. The process produced a large number of dark brown spots on the surface of the kelp, which looked like burnt marks and these marks were the direct cause of the low sensory evaluation at 50 °C. Meanwhile, the color score of the S−HP groups was lower because the ultraviolet irradiation in sunlight could accelerate chlorophyll destruction through free radical oxidation pathways [34]. Furthermore, long-time solar irradiation can also strengthen chlorophyll photosensitive oxidation degradation and lead to colorless products [35]. This was the reason why the kelp edges turned yellow and white.

### 3.6. System Performance

Table 8 shows the results of the energy consumption, the COP, and the SMER under various drying conditions. The total drying energy ranged from 8.6−20.7 kW·h and the COP and SMER of the system increased as the temperature and irradiance increased because the drying time became shorter. This result is consistent with the study of Tunckal et al. [36] in terms of heat pump drying of banana slices. The average COP of the S−HP mode was 5.379, which was significantly higher than that of the HP mode (2.399). When the drying temperature was 30 °C, 40 °C and 50 °C, the COP values of S−HP mode were 60%, 176% and 77% higher than those of HP mode, respectively. This was because the two heat sources in the S−HP system provided more heat energy per unit of time, which significantly improved the drying efficiency, resulting in 33% more energy saving compared with the HP system. The minimum energy consumption and the maximum COP and SMER of the S−HP system were obtained at 40 °C. This may be because solar irradiation under the set conditions could achieve a temperature of about 40 °C in the drying chamber, but an additional heat pump is needed for energy supplementation to reach 50 °C [37]. In addition, due to the longer drying time and low drying rate at 30 °C, the energy consumption was relatively high. Hu et al. [10] studied the HP of kelp knots. In contrast, the SMER of the proposed SHPD was 128.53% higher than that of the traditional system (1.630 kg/kW·h), indicating that the design can effectively improve the energy utilization rate and reduce energy consumption. Wang et al. [38], Qui et al. [16] and Mohanraj et al. [17] carried out the solar-assisted heat pump drying of mango, radish and coconut. The system COP values were 3.69, 3.49 and 2.54, respectively. In contrast, the newly designed S−HP system COP was 84.6%, 94.8% and 167.7% higher than those of the reported system, respectively. It shows that the greenhouse-type SHPD consumes less energy and effectively improves the system efficiency in low-temperature drying.

In order to assess the economic efficiency of the proposed SHPD, the energy consumption and the electricity cost for S−HP, HP and hot air drying were calculated and compared (Table 9). The energy consumption of S−HP and HP was measured by the electric energy meters installed in the instrument, and the energy consumption of hot air drying was calculated following a previous study [39]. S−HP saved 45.2% and 33.3% more operating costs compared with HP and hot air drying, respectively. Moreover, the carbon dioxide emission of S−HP is 52.2% and 63.2% less than that of HP and hot air drying, respectively [40]. S−HP makes full use of solar heat energy, which greatly improves the economy of the instrument and reduces pollutant emissions.

### 3.7. Principal Component Analysis

Principal component analysis (PCA) was performed on all variables to further investigate the association between the system performance and kelp drying quality (Figure 7). The cumulative contribution of the first two principal components accounted for 76.4%, meaning that they explained 76.4% of the drying effects. PC1 mainly accounted for the system performance information. Compared with HP groups, S−HP groups had unparalleled energy-saving advantages and could effectively shorten the drying time. The S−HP 40 °C (S−HP ≥ 400 W/m^2^) group showed the best energy-saving performance. In addition, energy consumption, COP and SMER show a strong correlation with drying time (Figure 7c). PC2 mainly showed the texture and sensory properties of the dried kelp. The sensory and textural qualities of the S−HP groups and the HP groups were comparable to each other overall, and the sensory qualities of the S−HP 40 °C and HP 40 °C groups were the best.

## 4. Conclusions

In order to solve the problems of high energy consumption, low efficiency and poor quality of the dried products during kelp drying, a novel greenhouse solar-assisted heat pump drying system with double evaporators was designed and manufactured, and its system performance was tested. As the drying temperatures and the irradiances increased within the ranges of 30−50 °C and 100−700 W/m^2^, respectively, the drying time was shortened, and the drying rate of S-HP was significantly higher than that of HP. The Page model was established to predict the moisture ratio of kelp. Considering the drying efficiency and quality of kelp, the optimal drying condition was a temperature of 40 °C and an irradiance above 400 W/m^2^ in S−HP mode. Under this condition, the drying time was 3 h, and the kelp retained uniform green in color, complete shape, a natural flavor and abundant mannitol. In addition, the COP was 6.810, the SMER was 3.725 kg/kWh, and the energy consumption was 45.2% lower than that of the HP mode.

## Figures and Tables

**Figure 1 foods-11-03509-f001:**
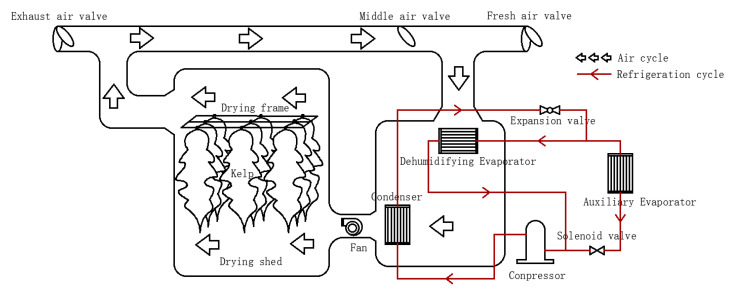
Schematic of the proposed SHPD.

**Figure 2 foods-11-03509-f002:**
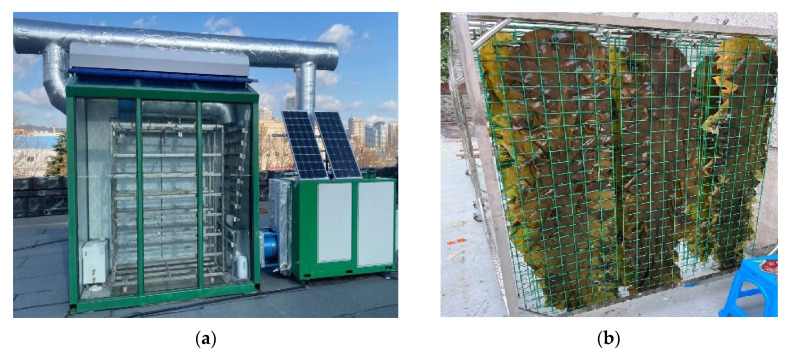
Photos of the proposed SHPD (**a**) and the geometric arrangement of the kelp for drying (**b**).

**Figure 3 foods-11-03509-f003:**
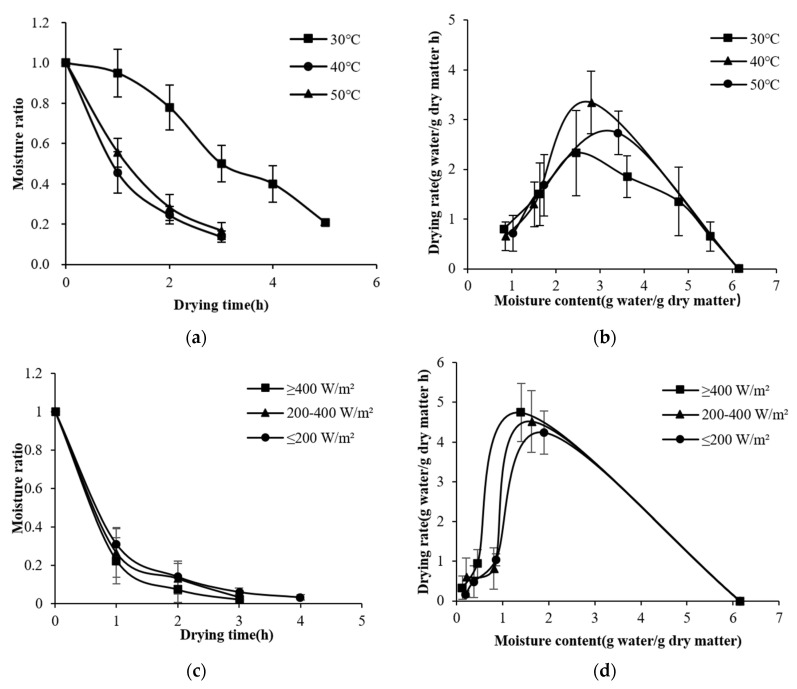
Moisture ratio (**a**) and drying rate (**b**) of kelp S−HP drying at different drying temperatures. Moisture ratio (**c**) and drying rate (**d**) of kelp S−HP drying under different irradiances. Irradiance curves under different drying conditions (**e**,**f**).

**Figure 4 foods-11-03509-f004:**
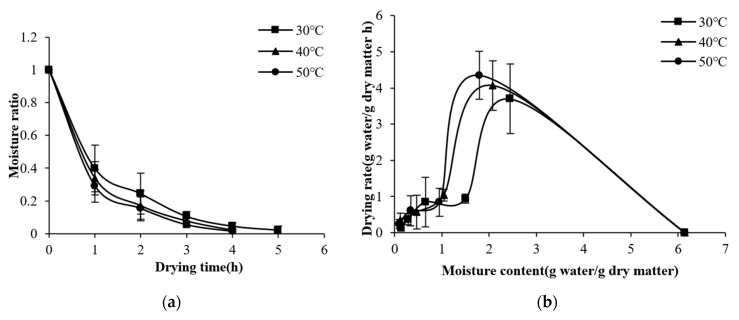
Moisture ratio (**a**) and drying rate (**b**) of kelp HP at different drying temperatures.

**Figure 5 foods-11-03509-f005:**
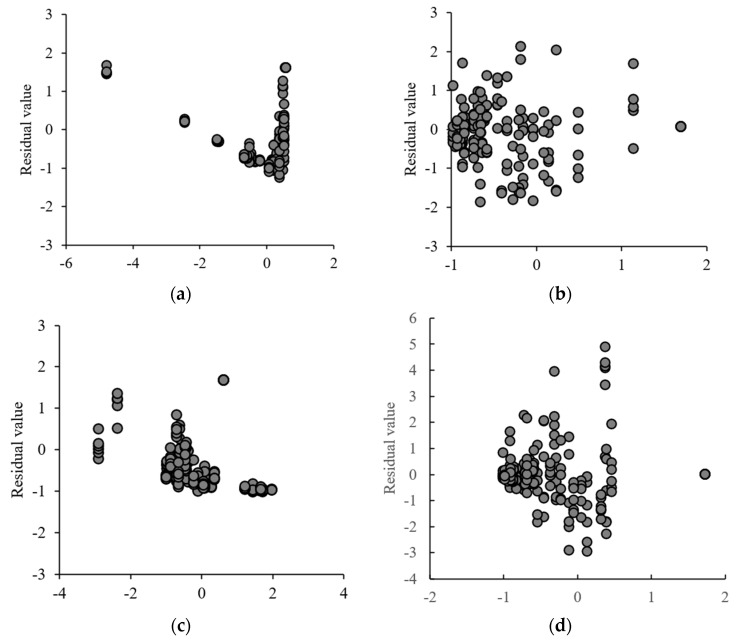
Residual plots of the Henderson−Pabis model (**a**), the Page model (**b**), the Wang and Sing model (**c**), the Lewis model (**d**) for all groups.

**Figure 6 foods-11-03509-f006:**
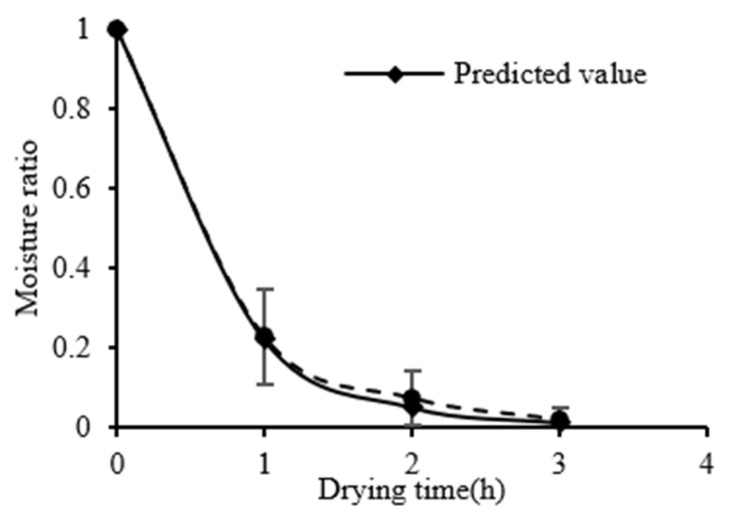
Validation of the drying model.

**Figure 7 foods-11-03509-f007:**
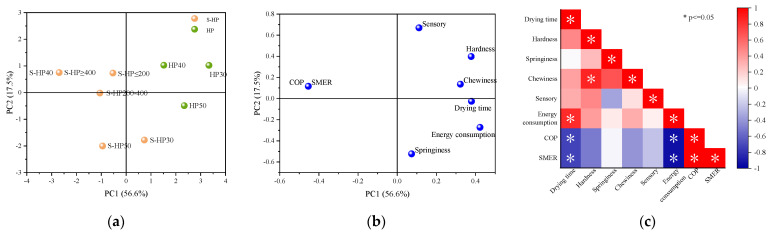
Score plot of principal components PC1 and PC2 (**a**) an, component plot in rotated space from the principal component analysis (**b**) and correlation plot (**c**).

**Table 1 foods-11-03509-t001:** Technical specifications of the equipment.

Component	Specifications	Model
Compressor	Cooling capacity: 12 kW; input power: 4.5 kW; exhaust volume: 14.4	ZW61KBC-TFP-522
Pump heat evaporator	Heat transfer area: 14.93 m^2^; air resistance loss: 95.51 Pa	Plate heat exchanger
Dehumidifying evaporator	Heat transfer area: 14.93 m^2^, air resistance loss: 95.51 Pa	Plate heat exchanger
Main condenser	Heat transfer area: 4.23 m^2^, air resistance loss: 94.51 Pa.	Plate heat exchanger
Thermal expansion valve	Nominal capacity: 3.98 kW.	SHF-20S-56-01
Circulation fan	Volume flow rate: 3000–5200 m^3^/h	GKF/F3.5D-2
Dehumidifying Fan	Volume flow rate of 3000–5200 m^3^/h	GKF/F3.5D-2
Refrigerant	R134a	

**Table 2 foods-11-03509-t002:** Arrangement of the drying tests for kelp.

Group	Drying Mode	Temperature (°C)	Irradiance (W/m^2^)
1	S−HP	40	≥400
2	S−HP	40	200–400
3	S−HP	40	≤200
4	S−HP	30	400–500
5	S−HP	40	
6	S−HP	50	
7	HP	30	none
8	HP	40	
9	HP	50	

**Table 3 foods-11-03509-t003:** Mathematical models for the data processing of kelp drying tests.

Model Name	Model	After Linearization	Reference
Henderson-Pabis	MR=Aexp(−Kt)	−lnMR=−lnA+kt	[19]
Page	MR=exp(−Ktn)	ln[−ln(MR)]=lnk+nlnt	[20]
Wang and Sing	$MR=1+At+bt^{2}$	$-lnMR=-ln\left(1+At+bt^{2}\right)$	[21]
Lewis	MR=exp(−kt)	−lnMR=kt	[22]

**Table 4 foods-11-03509-t004:** Sensory evaluation criteria for the dried kelp.

Factors	Evaluation Index	Score
Color	Uniform bright green	20~25
	Uniform dark brown	10~19
	Uneven light brown	0~9
Shape	Mannitol is precipitated on the surface of the kelp, without yellow/white edges	20~25
	No mannitol, with few yellow/white edges	10~19
	No mannitol, with many yellow/white edges	0~9
Flavor	Rich kelp flavor	20~25
	Light kelp flavor with undesired odor	10~19
	Obvious odor	0~9
Shrinkage	Slight shrinkage	20~25
	Severe shrinkage	10~19
	Shriveled and incomplete	0~9

**Table 5 foods-11-03509-t005:** Curve fitting parameters under different drying conditions.

Model Name	Drying Condition	R^2^	RMSE	χ^2^
Henderson−Pabis	S−HP ≥ 400 W/m^2^	0.999	0.795	3.790
	S−HP 200–400 W/m^2^	0.914	1.051	6.630
	S−HP ≤ 200 W/m^2^	0.890	4.463	119.504
	S−HP 30 °C	0.947	0.626	2.352
	S−HP 40 °C	0.999	0.795	3.790
	S−HP 50 °C	0.990	1.062	6.770
	HP 30 °C	0.992	0.849	4.323
	HP 40 °C	0.977	1.416	12.024
	HP 50 °C	0.924	3.71	82.583
Page	S−HP ≥ 400 W/m^2^	0.999	0.063	0.024
	S−HP 200–400 W/m^2^	0.876	0.086	0.044
	S−HP ≤ 200 W/m^2^	0.900	0.074	0.033
	S−HP 30 °C	0.939	0.049	0.014
	S−HP 40 °C	0.999	0.063	0.024
	S−HP 50 °C	0.974	0.056	0.019
	HP 30 °C	0.972	0.028	0.005
	HP 40 °C	0.959	0.033	0.007
	HP 50 °C	0.873	0.059	0.021
Wang and Sing	S−HP ≥ 400 W/m^2^	0.917	0.142	0.121
	S−HP 200–400 W/m^2^	0.771	0.204	0.249
	S−HP ≤ 200 W/m^2^	0.726	0.228	0.313
	S−HP 30 °C	0.903	0.085	0.044
	S−HP 40 °C	0.917	0.142	0.121
	S−HP 50 °C	0.884	0.105	0.067
	HP 30 °C	0.774	0.253	0.384
	HP 40 °C	0.776	0.209	0.263
	HP 50 °C	0.739	0.224	0.301
Lewis	S−HP ≥ 400 W/m^2^	0.767	0.079	0.038
	S−HP 200–400 W/m^2^	0.914	0.045	0.012
	S−HP ≤ 200 W/m^2^	0.890	0.054	0.018
	S−HP 30 °C	0.903	0.044	0.012
	S−HP 40 °C	0.950	0.079	0.038
	S−HP 50 °C	0.948	0.030	0.005
	HP 30 °C	0.992	0.095	0.014
	HP 40 °C	0.977	0.025	0.037
	HP 50 °C	0.924	0.046	0.013

**Table 6 foods-11-03509-t006:** Effective moisture diffusivity under different drying conditions.

Drying Condition	Linear Regression Fitting Formula	*D_eff_* (10^−11^m^2^/s)	R^2^
S−HP ≥ 400 W/m^2^	y = −4.358 × 10^−4^x + 0.013	11.316	0.999
S−HP 200–400 W/m^2^	y = −3.053 × 10^−4^x + 0.104	7.926	0.914
S−HP ≤ 200 W/m^2^	y = −2.411 × 10^−4^x + 0.212	6.260	0.890
S−HP 30 °C	y = −0.936 × 10^−4^x − 0.207	5.431	0.947
S−HP 40 °C	y = −4.358 × 10^−4^x + 0.013	11.316	0.999
S−HP 50 °C	y = −1.706 × 10^−4^x + 0.614	10.428	0.990
HP 30 °C	y = −2.086 × 10^−4^x − 0.031	1.037	0.992
HP 40 °C	y = −2.606 × 10^−4^x − 0.002	1.295	0.977
HP 50 °C	y = −2.881 × 10^−4^x − 0.020	1.432	0.924

**Table 7 foods-11-03509-t007:** Textural indices under different drying conditions.

Drying Condition	Hardness (N)	Springiness (mm)	Chewiness (mJ)	Sensory Score
S−HP ≥ 400 W/m^2^	104.61 ± 19.79 ^cd^	0.65 ± 0.38 ^a^	61.01 ± 45.776 ^abc^	79.80 ± 8.19 ^ab^
S−HP 200–400 W/m^2^	148.31 ± 25.04 ^bc^	0.75 ± 0.08 ^a^	80.12 ± 61.24 ^abc^	77.47 ± 4.41 ^ab^
S−HP ≤ 200 W/m^2^	141.93 ± 42.05 ^bc^	0.58 ± 0.54 ^a^	88.12 ± 31.47 ^abc^	75.53 ± 4.67 ^ab^
S−HP30 °C	103.86 ± 30.07 ^cd^	0.71 ± 0.08 ^a^	59.25 ± 19.83 ^abc^	73.67 ± 7.44 ^ab^
S−HP40 °C	104.61 ± 19.79 ^cd^	0.65 ± 0.38 ^a^	61.01 ± 45.776 ^abc^	79.80 ± 8.19 ^ab^
S−HP50 °C	103.88 ± 45.33 ^cd^	0.73 ± 0.10 ^a^	68.80 ± 40.82 ^abc^	69.93 ± 9.90 ^b^
HP30 °C	204.17 ± 59.74 ^a^	0.62 ± 0.57 ^a^	113.07 ± 102.55 ^ab^	79.67 ± 2.20 ^ab^
HP40 °C	160.11 ± 55.65 ^ab^	0.65 ± 0.11 ^a^	63.01 ± 36.22 ^abc^	84.80 ± 2.31 ^a^
HP50 °C	164.53 ± 25.37 ^ab^	0.82 ± 0.03 ^a^	117.76 ± 17.51 ^a^	80.13 ± 6.02 ^ab^
Raw	62.51 ± 51.62 ^d^	0.62 ± 0.12 ^a^	35.23 ± 33.79 ^bc^	77.37 ± 2.85 ^ab^

Different letters (^a–d^) indicate significant differences between groups as determined via one-way ANOVA (*p* < 0.05).

**Table 8 foods-11-03509-t008:** System energy consumption, COP and SMER under different drying conditions.

Drying Condition	Energy Consumption (kW·h)	COP	SMER (kg/kW·h)
S−HP ≥ 400 W/m^2^	8.6	6.810	3.725
S−HP 200–400 W/m^2^	10.9	5.486	2.876
S−HP ≤ 200 W/m^2^	11.9	5.180	2.679
S−HP 30 °C	20.3	3.590	1.660
S−HP 40 °C	8.6	6.810	3.725
S−HP 50 °C	15.5	4.400	2.179
HP 30 °C	20.7	2.244	0.798
HP 40 °C	18.0	2.463	0.938
HP 50 °C	17.6	2.491	0.956

**Table 9 foods-11-03509-t009:** Energy consumption of different drying methods.

Drying Method	Energy Consumption (kW·h/kg)	Electricity Cost (RMB)	CO_2_ Emission (g/kg)
S−HP	0.52	0.46	428.71
HP	0.95	0.84	897.30
Hot air drying	1.17	1.04	1166.49

The electricity cost was calculated based on the current charge of 0.89 yuan/kW·h.

## Data Availability

The data showed in this study are contained within the article.

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
