# Peer review of "Design of a Greenhouse Solar-Assisted Heat Pump Dryer for Kelp (Laminaria japonica): System Performance and Drying Kinetics"

_foods, 2022, doi:10.3390/foods11213509_

Round 1

Reviewer 1 Report

The submitted article is a continuation of previously published research-Heat Pump Drying of Kelp (Laminaria japonica): Drying Kinetics and Thermodynamic Properties. In this article, a greenhouse-type double evaporator SHPD was designed to address the batch drying and processing needs of kelp and other bulk aquatic products and to circumvent the typical heat and humidity mismatch. The drying performance and kinetics of the kelp were examined, and theoretical and technical support was provided for the development of energy-saving, high-efficiency drying technology and equipment for bulk, low-value aquatic products. The article is nicely presented. The text was somehow easy to follow. However, I have noticed some minor issues that need the Authors attention:

Page 5, line 140, why did the authors choose Wang and Sing model for kelp drying and not the Lewis model like in the previous article?

Page 6, line 164, The Principal Component Analysis and the used program should be mentioned in this section,

Page 10, please, provide R2 in table 6,

Page 11, please write letters that show significantly different means of observed data in superscript and explain the meaning below Table 7,

Page 12, consider adding a color correlation diagram before PCA or just provide correlation values.

Reference - The format of references in the reference list should be standardized according to the requirement of Foods (refer to the Guide of Authors).

Reviewer 2 Report

The manuscript is interesting, but I would like to ask about the novelty if you compared it with below article which published in processes

Heat Pump Drying of Kelp (Laminaria japonica): Drying Kinetics and Thermodynamic Properties

In general, there are many cites in the manuscript that doesn't contain spaces between words

Line 110: there is no space between is and 10 ℃

Line 118-119: no spaces between measuring unit and the values

Line 198-199: there are no spaces between some words

Line 230-231: the phrase is not clear “The Deff of kelp drying varied from 1.037×10-11 m2/s to 11.316×1011m2/s, within reasonable bounds for biological materials”

258-260: how you explain the changes in color (brighter), did you measure something or did you measure mannitol precipitation

Table 7. contains letters in normal case, I think it should be superscript letters and p- values should be added

Reviewer 3 Report

Authors in the article entitled "Design of Greenhouse Solar Assisted Heat Pump Dryer of Kelp 2 (Laminaria japonica): System Performance and Drying Kinetics" presented currently interesting information. This paper fit into the scientific scope of the FOODS Journal and above all in the thematic special issue.

The introduction is more detailed, includes 14 references and I have no major revisions to it. Perhaps I have to note for some typos and errors that indicate lack of control when sending the manuscript. These are missing or redundant spaces (e.g. lines 41, 47, ...), substitutions of commas instead of periods (e.g. line no. 38), missing periods after brackets (e.g. line 47), etc. So it is necessary to proofread the entire text.

In the material and methodology chapter, part of the text is devoted to the S-HP drying system. It would be necessary to refer to the literature in the sense of its comparison, evaluation, use.

In Table 1, the units are m2 (correctly with the index - m2). Here, I recommend proofreading the text in some names/words, too.

I have reservations about part of the text:

"A centesimal system was adopted as shown in Table 4, and 5 pieces of kelp were sampled from each group. 5 professionals trained in food sensory evaluation scored each group of kelp, and the average value was calculated as the final score."

A centesimal system - references?

5 professionals trained - evaluators? with certificate, training? (type of training, test or course?, frequency of evaluation?, type of samples they evaluate?, gender or age?)

It is quite a shame that color measurement was not used - CIELAB colorimeter/spectrophotometer (CIE L*a*b*/hunter Lab). Therefore, I think that more detailed information is needed regarding the evaluation of color in a subjective way. How was a repeat attempt ensured - repetition of the experiment?

There are about 14 references in the chapter Result and discussion, which would be enough if the authors handled them better. However, it is my recommendation that in each sub-chapter of this important part of the Article there is a citation and the results of the authors' work are discussed.

Again, there are several typos in this chapter and the authors need to concentrate on removing them. However, the tables, figures and main passages meet the requirements for this type of work.

In the conclusion chapter, I recommend not to repeat the aforementioned theses, but to focus on the summary, recommendations and evaluation of the entire experiment.

In the section dedicated to references, is needed to edit the citations according to the pattern.

Reviewer 4 Report

The manuscript "Design of Greenhouse Solar Assisted Heat Pump Dryer of Kelp (Laminaria japonica): System Performance and Drying Kinetics" is very well elaborated. The preparation of the experiment, methodology and carrying out of the experiment are carried out very reliably. The description and discussion of the obtained results is logical and supported by statistical analysis. The use of rounded lines in Figures 3, 4, 6 would be an improvement, but this does not significantly affect the overall evaluation of the manuscript.
